# Leisure Time Physical Activity and SARS-CoV-2 Infection among ELSA-Brasil Participants

**DOI:** 10.3390/ijerph192114155

**Published:** 2022-10-29

**Authors:** Francisco José Gondim Pitanga, Maria da Conceição Almeida, Bruce B. Duncan, José Geraldo Mill, Luana Giatti, Maria del Carmen B. Molina, Maria de Jesus Mendes da Fonseca, Maria Inês Schmidt, Rosane Harter Griep, Sandhi Maria Barreto, Sheila Maria Alvim de Matos

**Affiliations:** 1Postgraduate Program in Rehabilitation Sciences, Institute of Health Sciences, Federal University of Bahia (UFBA), Salvador 40110-150, Brazil; 2Gonçalo Moniz Institute, Oswaldo Cruz Foundation, Salvador 40296-710, Brazil; 3Postgraduate Program in Epidemiology, Federal University of Rio Grande do Sul, Porto Alegre 90035-003, Brazil; 4Department of Physiological Sciences, Federal University of Espírito Santo, Vitória 29075-910, Brazil; 5Research Group on Epidemiology on Chronic and Occupational Diseases (GERMINAL), Faculty of Medicine & Clinical Hospital, Federal University of Minas Gerais, Belo Horizonte 31270-901, Brazil; 6Department of Nutrition, Federal University of Espírito Santo, Vitória 29075-910, Brazil; 7National School of Public Health, Oswaldo Cruz Foundation, Rio de Janeiro 21049-900, Brazil; 8Laboratory of Health and Environment Education, Oswaldo Cruz Institute, Oswaldo Cruz Foundation, Rio de Janeiro 21049-900, Brazil; 9Institute of Collective Health, Federal University of Bahia, Salvador 40110-040, Brazil

**Keywords:** physical activity, SARS-CoV-2, COVID-19

## Abstract

The regular practice of physical activity (PA) can reduce the chance of aggravation of the disease and lower rates of hospitalization and mortality from COVID-19, but few studies have analyzed the association of PA with the risk of infection by SARS-CoV-2. The aim of the study was to analyze the association between PA and self-reported SARS-CoV-2 infection. A longitudinal study was conducted with data from 4476 ELSA-Brasil participants who had their PA analyzed twice, once in 2016–2018 and again in 2020. PA was identified using the IPAQ at both follow-up moments and categorized into four groups: (a) remained physically inactive (reference); (b) remained physically active; (c) became physically active in the second moment; and (d) became physically inactive in the second moment. The variables of age, sex, obesity, hypertension, diabetes and specific protective practices against COVID-19 were tested as possible confounders. Data were analyzed by logistic regression. A 95% confidence interval (CI) was used. Remaining physically active was associated with a 43% reduction in the risk of SARS-CoV-2 infection only among those who used specific practices to protect against COVID-19, OR = 0.57 and CI = 0.32-0.99. The results suggested that regular practice of PA can reduce the risk of SARS-CoV-2 infection, especially among those who have used specific practices to protect against COVID-19 during the pandemic.

## 1. Introduction

Since the beginning of the COVID-19 pandemic in March 2020 [1], several researchers have been trying to identify interventions that could mitigate the adverse effects of the disease. Among them, we can mention: vaccines, medicines, prevention measures, protection and sanitary control and behavioral changes (diet, physical activity), among others.

Previous studies on influenza and H1N1 showed that the possibility of worsening clinical status and mortality was lower in more physically active people [2,3]. It was then speculated that these findings could be transported to COVID-19 since it was an infection with similar viral characteristics [4].

As of 2021, specific articles on physical activity and COVID-19 began to be published, demonstrating that among the most physically active individuals, there was a lower chance of aggravation of the disease and lower rates of hospitalization and mortality from COVID-19 [5,6,7]. The scientific evidence produced led the United States Center for Disease Control and Prevention (CDC) to consider physical inactivity as a high-risk factor for the development of the most serious forms of COVID-19 [8].

Despite all this evidence, few studies have investigated whether physical activity can prevent community infectious processes, [9] including the SARS-CoV-2 virus [10,11], a precursor of COVID-19 [12]. A systematic review and meta-analysis study showed that the risk of community-acquired infectious diseases was lower in more physically active people [9]. The authors suggested that this fact could be attributed to the increase in the strength of the mucosal immune barrier (salivary IgA immunoglobulin) and a higher concentration of immune cells, such as T lymphocytes [9,13].

It is highlighted that individual and social protection measures, such as hand washing, use of alcohol gel, use of a mask and physical distancing, are also important actions that can reduce the process of spreading SARS-CoV-2 [14]. It is important to note that these measures must be conducted both in daily activities and during the practice of physical activity in outdoor environments and physical conditioning centers as long as they are open during the pandemic.

Specifically, with regard to the influence of physical activity on the infectious process caused by SARS-CoV-2, two recent studies showed that those who were more physically active were less infected when compared with those who were less physically active [10,11]. Studies of this nature can make important contributions to public health since making the population more physically active can mitigate the adverse effects of both the current and future pandemics with similar characteristics.

Thus, the main objective of the present study was to analyze the association between leisure-time physical activity (LTPA) and self-reported SARS-CoV-2 infection in Longitudinal Study of Adult Health (ELSA-Brasil) participants.

## 2. Materials and Methods

### 2.1. Population and Sample

The ELSA-Brasil is a cohort that includes 15,105 active or retired public servants aged between 35 and 74 years, with an average of 52.1 ± 9.1 years at the baseline (2008–2010), from six higher education institutions located in the cities of Salvador, Vitória, Belo Horizonte, Rio de Janeiro, São Paulo and Porto Alegre, belonging to three Brazilian regions (northeast, southeast and south), with well-defined cultural and socio-demographic differences. Tourism is one of the most important activities in the economy of the northeast region and the dry climate is one of the obstacles to development in the region. The southeast region has the largest gross domestic product (GDP) in Brazil and the economy is based on the industrial, financial and commercial sectors. The southern region has an economy based on plant extraction and agriculture. This region has the second-highest GDP in Brazil.

Although the study methodology has been previously described, its main objective was to contribute relevant information with respect to the development and progression of clinical and subclinical chronic diseases, particularly cardiovascular events and diabetes. During the follow-up, the participants answered a questionnaire about their general health conditions, family history of diseases, medication use and mental health, among other topics. In addition, a series of laboratory and physical examinations were performed [15,16].

Between 2012 and 2018, two face-to-face follow-ups were conducted with cohort participants and an additional follow-up with online interviews during the first year of the COVID-19 pandemic (COVID Stage). It is noteworthy that the COVID Stage was not foreseen in the objectives and initial design of the study, but with the emergence of the pandemic in 2020, it was decided to collect information about the disease.

For the present study, with a longitudinal observational design (participants were followed for a certain period of time), those who attended the second follow-up (2016–2018, with about an 86.7% participation rate) and in the COVID Stage of ELSA-Brasil (2020) were considered, with complete data on the variables involved in the process of analysis, totaling 4476 participants (2595 women) from five of the six participating centers, with the exception of São Paulo.

At the baseline and at each follow-up stage, the ELSA-Brasil was approved by all the Research Ethics Committees of the investigation centers involved. All participants signed the free and informed consent form, guaranteeing the secrecy and confidentiality of the data.

### 2.2. Data Production

Data were collected by a team of evaluators and interviewers trained and certified by a quality control committee capable of executing the study protocol at any ELSA-Brasil Research Center [16]. Specifically, regarding the COVID Stage, data were mostly collected through an online questionnaire sent to participants through a specific link.

#### 2.2.1. Physical Activity Assessment

To identify and quantify physical activity, the International Physical Activity Questionnaire (IPAQ), long version, was used, which consists of questions related to the frequency and duration of physical activities (walking, moderate physical activity and vigorous physical activity) performed at work, while commuting, domestic activities and leisure time [17]. In the ELSA-Brasil, only the domains of leisure time and commuting were evaluated, but in the COVID Stage, the domain of domestic activities was also evaluated. Physical activity was measured in minutes per week by multiplying the weekly frequency by the duration of each of the activities performed. For the purposes of this study, only LTPA was used, with the following categorization: 0 = physically inactive (from 0 to less than 150 min of walking or moderate physical activity per week or from 0 to less than 60 min of vigorous activity per week or 0 to less than 150 min per week of any combination of walking, moderate or vigorous physical activity) and 1 = physically active (≥150 min of walking or moderate activity per week or ≥60 min of vigorous activity per week or ≥150 min per week of any combination of walking, moderate or vigorous physical activity). Subsequently, LTPA was categorized into four groups, considering the two follow-up moments: a) remained physically inactive (reference); b) remained physically active; c) became physically active in the second moment; and d) became physically inactive in the second moment.

#### 2.2.2. Assessment of Self-Reported SARV-CoV-2 Infection

SARV-CoV-2 infection was assessed by the self-reported criterion positive test for COVID-19 through the following question: Did you have a positive test for COVID-19?

#### 2.2.3. Evaluation of Covariates in the Second Follow-Up

##### Obesity

Obesity was identified through the body mass index (BMI), applying the equation BMI = weight (kg)/height(m)^2^, adopting the following cut-off point; obesity = 0 if BMI < 30.0 and obesity = 1 if BMI ≥ 30.0. Body weight was obtained in the second follow-up of the cohort (2016–2018), in the morning, after 8 to 12 h of fasting and with the participant without shoes and wearing light clothing. A Toledo^®®®^ electronic scale was used, with a capacity of up to 200 kg. To measure height, a SECA^®®®^ stadiometer was used, with the participant positioned standing up and strictly following the Frankfurt plane.

##### Hypertension

Data on blood pressure was obtained in the second follow-up (2016–2018) and was taken using a validated oscillometric device (Omron HEM 705CPINT) after a 5 min resting period, with the subject in a seated position, in a quiet, temperature-controlled room (20–24 °C). Three measurements were taken at one-minute intervals. The mean of the last two measurements was used for the analyses. A participant was classified as hypertensive if systolic blood pressure was ≥ 140 mmHg, if diastolic blood pressure was ≥ 90 mmHg, or if they had taken any medication to treat hypertension in the preceding two weeks.

##### Diabetes

Blood samples were also obtained in the second follow-up (2016–2018) and were collected after a 12 h fast, stored in a freezer at −80 °C and sent to the certified central laboratory in São Paulo. An oral glucose tolerance test (OGTT) was administered to all participants without a known diabetes diagnosis [18]. Glycaemia was measured using the enzymatic colorimetric method (ADVIA 1200 Siemens, Deerfield, IL, USA), and glycated hemoglobin A1c was measured using high-pressure chromatography (HPLC-Bio-Rad Laboratories, Hercules, CA, USA).

Diabetes was defined as A1c ≥ 6.5% (48 mmol/mol), fasting glycemia≥ 126 mg/dL (7.0 mmol/L) or OGTT ≥  200 mg/dL (11.1 mmol/L), according to the American Diabetes Association (ADA) criteria; by insulin and antidiabetic drug use; or by the self-reported medical diagnosis of diabetes.

#### 2.2.4. COVID Stage Covariates

##### Specific Practices to Protect against COVID-19

The specific practices of protection against COVID-19 were obtained through questions referring to three variables collected in the COVID Stage, considering a previous publication that suggested reduction of the spread of the SARS-CoV-2 through these procedures [14]: use of 70% gel or liquid alcohol on the hands, washing hands with soap and water for 20 s and wearing a mask whenever leaving the house. Those who responded that they “always” had this behavior, considering the three variables at the same time, were categorized as 0, while those who responded to any combination of “almost always”, “sometimes”, “rarely” or “never” were categorized as 1.

### 2.3. Data Analysis

Descriptive measures (proportions) were calculated for all categorized variables. Differences were identified using Pearson’s chi-square test. The associations between the dependent variable (SARS-CoV-2 infection identified by the self-reported criterion of a positive test for COVID-19) and the dependent variable (LTPA) were evaluated by means of logistic regression, to estimate the odds ratio (OR), with 95% confidence interval. The following variables were considered as potential confounders: age, sex, obesity, hypertension, diabetes and specific protective practices against COVID-19. The analysis for confounding was performed by comparing the OR of the association between LTPA and SARS-CoV-2 infection starting from the complete model and removing each of the possible confounding variables. [19]. The parameter used to identify the difference between the associations was 10%. The analysis of effect modification was performed by examining stratum-specific punctual measurements and their confidence intervals. If the punctual measurement of a factor in a specific stratum was not in the confidence interval of another factor in the same stratum, that indicated an effect modification. In the modeling process, no confounding or effect modification variables were identified. In addition, considering that specific protective measures against COVID-19 seem to be an important behavior for the prevention of SARS-CoV-2 infection, we chose to consider them as an effect modification, a priori [14]. Therefore, the best model to analyze the association between LTPA and SARV-CoV-2 infection was the one stratified by specific measures to protect against COVID-19.

## 3. Results

A total of 1881 men (41.9%) and 2595 women (58.1%) were included in the analysis. It was observed that men were more hypertensive and diabetic, while women were more obese. Regarding LTPA, men remained more active during the follow-up. As for the specific measures to protect against COVID-19, women used these procedures more. No statistically significant differences were observed between men and women in age, as well as in the prevalence of self-reported SARS-CoV-2 infection (Table 1).

Associations between LTPA and self-reported SARS-CoV-2 infection were different between those who did or did not adopt the recommendations for protection against COVID-19 (Table 2). In those who followed the specific protection practices against COVID-19, also complying with the recommendations for physical activity (before and during the pandemic) was associated with a 43% reduction in the risk of being infected with SARS-CoV-2 (OR = 0.57; 95%CI 0.32–0.99). In those who did not comply with the recommendations for physical activity before the pandemic but became active during the follow-up, protection was lower and not statistically significant (OR = 0.64; 95%CI 0.31–1.30). In those who practiced physical activity before the pandemic but did not maintain it during the follow-up, the association was even lower (OR = 0.88; 95%CI 0.50–1.55). No associations were observed among those who did not follow COVID-19 protection recommendations, even in those who remained physically active.

## 4. Discussion

The study longitudinally analyzed the association between LTPA and self-reported SARS-CoV-2 infection. It was observed that those who remained physically active during the follow-up and conducted specific practices to protect against COVID-19 during the pandemic were less infected when compared with those who remained physically inactive.

One of the first systematic review studies with meta-analysis that examined the association between habitual physical activity and the risk of infectious diseases showed that a higher level of habitual physical activity is associated with a 31% risk reduction of community-acquired infectious disease, as well as a 37% reduction in the risk of mortality from infectious diseases [9], leading the authors to conclude that regular, moderate to vigorous physical activity is associated with a reduced risk of community-acquired infectious diseases and infectious disease mortality. It is important to note that in this study, SARS-CoV-2 infection was not evaluated.

Specifically, regarding physical activity as a protective factor for SARS-CoV-2 infection observed in our study, it was also seen in two recent publications. In the first one, with data from 12,500 people randomly selected from the Korea National Health Insurance Service database [10], the authors observed that higher levels of regular physical activity were associated with a lower risk of infection and mortality from COVID-19, highlighting the importance of maintaining adequate levels of physical activity alongside social distancing amid the COVID-19 pandemic.

In another study, also conducted in South Korea, with the participation of 212,768 adults, it was observed that those who participated in aerobic physical activity and muscle strengthening, especially at moderate intensity, were less likely to be infected by SARS-CoV-2, as well experience severe COVID-19 illness or COVID-19-related death. Furthermore, the recommended range of metabolic equivalent tasks (METs; 500–1000 METs min per week), i.e., moderate physical activity, was associated with the maximum beneficial effect size for a reduced risk of SARS-CoV-2 infection, severe COVID-19 illness or COVID-19-related death [11]. Based on these results, the study authors suggested that public health policies and strategies to promote physical activity in the population could reduce the risk of SARS-CoV-2 infection and minimize adverse consequences in patients with COVID-19.

More recently, in 2022, a systematic review with meta-analysis concluded that physical activity could reduce the risk of SARS-CoV-2 infection, in addition to reducing the likelihood of adverse effects of COVID-19, such as hospitalizations, severity of illness and mortality. They also emphasized that due to the limitations of the studies, the results needed to be interpreted with caution [20].

Regarding specific protection practices against COVID-19 and corroborating our results, a recent systematic review with meta-analysis showed that several individuals and social protection measures, including hand washing, mask use and physical distancing, were associated with reductions in the incidence of COVID-19 [14]. In this same study, the authors suggested that efforts to implement public health measures should consider community health and sociocultural needs and that future research is needed to better understand the effectiveness of these measures in the context of vaccination against COVID-19.

The results found in our study on the reduction of the risk of SARS-CoV-2 infection in physically active individuals can be partially attributed to a higher concentration of immune cells, such as T lymphocytes, in addition to the increase in the strength of the mucosal immune barrier (salivary IgA immunoglobulin) [9,13,21], as it has the potential to provide greater immunity against different types of viruses and bacteria that enter the human body through the oral cavity and upper airways [22] (Figure 1).

Regarding the association between physical activity and salivary IgA immunoglobulin, some studies showed that physical exercise could cause increases in its concentration. In one of them, the authors subjected 9 people to 12 weeks of moderate physical training, compared with 10 people who did not exercise. At the end of the experiment, they observed a significant increase in salivary IgA immunoglobulin in the exercise group after training, leading the authors to conclude that regular and moderate exercise results in increased resting in salivary IgA immunoglobulin, which may contribute to a decreased risk of infection [13].

In another study conducted with elderly men and women in order to analyze the influence of age and sex on salivary IgA immunoglobulin in response to moderate physical training, the authors demonstrated that an increase in mucosal immune function (salivary IgA immunoglobulin) after regular moderate exercise training occurred in elderly people aged between 60 and over 70 years, both in men and women [21].

The study’s main strength was being one of the first to show the possible association of physical activity as a protective factor for SARS-CoV-2 infection. In addition, this was a cohort that followed a large number of civil servants from six higher education institutions in Brazil, which, despite not reflecting the entire Brazilian population, was composed of people of both sexes from three regions of Brazil with well-defined cultural and socio-demographic differences, such as ethnic, racial, economic and educational aspects. On the other hand, the fact that the information used for the diagnosis of the infection was indirectly obtained, in addition to the small number of self-reports of positive tests for COVID-19, constituted the main limitations of the study. Another possible limitation of the study was that the information on physical activity was obtained by self-reported questionnaires, which nevertheless are widely used instruments in national and international studies.

## 5. Conclusions

The results obtained based on the present study suggest that the regular practice of leisure time physical activity, together with specific practices to protect against COVID-19 (use of alcohol gel or liquid 70% on the hands, washing hands with soap and water for 20 s and wearing a mask whenever you leave the house) may be a protective factor for SARS-CoV-2 infection and should be encouraged in the population to mitigate the adverse effects of SARS-CoV-2 infection, as well as future infectious processes with characteristics similar. In addition, the data support the maintenance of physical activity among non-pharmacological measures to protect against COVID-19, especially considering its protective effect on the severity of the disease. Further studies with similar characteristics and with the inclusion of a greater number of infected people, in addition to the analysis of the duration and intensity of physical exercise, are suggested to better clarify the potential of physical activity as a protective factor for SARS-CoV-2 infection.

## Figures and Tables

**Figure 1 ijerph-19-14155-f001:**
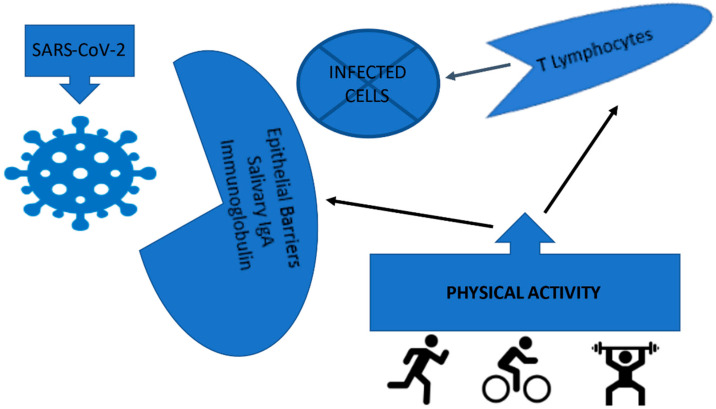
Effect of physical activity on the higher concentration of immune cells, such as T lymphocytes, in addition to the increase in the strength of the mucosal immune barrier (salivary IgA immunoglobulin).

**Table 1 ijerph-19-14155-t001:** Characteristics of the study population. Longitudinal Study of Adult Health ELSA-Brasil (2016–2018) and COVID Stage (2020).

	Men (*n* = 1.881)		Women (*n* = 2.595)		*p*-Value
	*n*	%	*n*	%	
Leisure time physical activity					
(Second follow-up–COVID Stage)					
Inactive-Inactive	535	28.4	1017	39.1	
Active-Active	729	38.7	740	28.5	
Inactive-Active	249	13.2	355	13.6	
Active-Inactive	368	19.5	483	18.6	<0.001
Self-reported SARS-CoV-2 infection					
(Positive COVID-19 Test)					
Yes	69	3.5	85	3.2	
No	1853	96.5	2581	96.8	0.456
AGE					
<60 years	998	51.9	1433	53.7	
≥60 years	924	48.1	1233	46.3	0.221
Hypertension					
Yes	841	45.5	1038	40.3	
No	1006	54.5	1535	59.7	<0.001
Diabetes					
Yes	272	16.9	310	12.7	
No	1334	83.1	2126	87.3	<0.001
Obesity					
Yes	393	20.9	671	25.9	
No	1481	79.1	1919	74.1	<0.001
Specific protection practices against COVID-19					
Always	959	50.2	1528	57.7	
Almost always, sometimes, rarely or never	953	49.8	1118	42.3	<0.001

**Table 2 ijerph-19-14155-t002:** Association between leisure-time physical activity and self-reported SARS-CoV-2 infection stratified by specific protective practices against COVID-19. Longitudinal Study of Adult Health ELSA-Brasil (2016–2018) and COVID Stage (2020).

Leisure Time Physical Activity (Second Follow-Up–COVID Stage)	Specific Protection Practices against COVID-19 (YES) OR (IC 95%)	Specific Protection Practices against COVID-19 (NO) OR (IC 95%)
Inactive-Inactive Active-Active Inactive-Active Active-Inactive	1 0.57 (0.32–0.99) 0.64 (0.31–1.30) 0.88 (0.50–1.55)	1 0.80 (0.45–1.42) 0.52 (0.20–1.36) 0.50 (0.22–1.17)

## Data Availability

The data presented in this study are openly available at http://elsabrasil.org.

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
