# Peer review of "Leisure Time Physical Activity and SARS-CoV-2 Infection among ELSA-Brasil Participants"

_ijerph, 2022, doi:10.3390/ijerph192114155_

Round 1

Reviewer 1 Report

The paper deals with an interesting topic wich has potenitial  for lowering the mortality rate and saving lifes cosed by Covid-19 infection. 

According to the aforementioned, it is my recommendation that the paper should be accepted for publishing in the proposed form.

Reviewer 2 Report

There are previous studies that review the subject of this manuscript in depth and with great methodological robustness, so the findings do not provide new information. In addition to the limitations that the authors describe in the study, they only consider physical activity during leisure time, which may be introducing important biases, especially if the physical activity at work, commuting, and domestic activities were also measured.

Reviewer 3 Report

The purpose of this study was to analyze the association between leisure time physical activity (LTPA) and self-reported SARS-CoV-2 infection in ELSA-Brasil participants.

The paper is misleading in the current form, I would recommend publication in case the authors agree to address the following issues:

1)    Introduction

Line 67-69 seems out of context, write it better, explain when these measures should be taken (if you go to the gym for example). In 2020 we were in full lockdown for COVID-19. 

2)      Materials and methods 

Line 99-100 enter the link to the online questionnaire and define the last access. The questionnaire could be added in the supplementary materials.

Line 106 mention the first follow-ups, do you mean the one from 2008-2010? give the date.

3)      Discussion

A large part of discussion is outside the results, please shorten this paragraph and rephrase some of the sentences.

4)      Conclusions

Please downsize the importance of the findings, it is too much to claim that physical activity reduces the risk of SARS-CoV-2 (it could be a protective factor for several diseases and also for SARS-CoV-2).

Good revision work and good luck

Reviewer 4 Report

Dear authors,

I am glad to contribute to the peer review of your manuscript. You have invested a lot of time and effort in carrying out this research and in writing this manuscript. I will now give you some advices and recommendations, based on my experience and expertise, to improve your research. Please, let me know if you have any doubt about my comments.

Brief summary

Main goals: to analyze the association between leisure time physical activity (LTPA) and self-reported SARS-CoV-2 infection in ELSA-Brasil participants.

Main contributions: The study provides insight on how LTPA and COVID-specific prevention behaviors interrelate to reduce risk of infection in adult population in Brasil.

Strengths: Representative sample, longitudinal data, deep and thorough discussion.

General comments

·        Is the manuscript clear, relevant for the field and presented in a well-structured manner?

I believe the manuscript is relevant for the field and its general structure is appropriate. However, the materials and methods section lacks of crucial information for understanding and replicating the study. Please see specific comments.

·        Are the cited references mostly recent publications (within the last 5 years) and relevant? Does it include an excessive number of self-citations?

The references are recent, but there is not any reference from 2022. There is not excessive self-citation.

·        Is the manuscript scientifically sound and is the experimental design appropriate to test the hypothesis?

Research design is not specified in the manuscript. There are no hypotheses, only a general goal.

·        Are the manuscript’s results reproducible based on the details given in the methods section?

No, the methods section lacks of fundamental information.

·        Are the figures/tables/images/schemes appropriate? Do they properly show the data? Are they easy to interpret and understand? Is the data interpreted appropriately and consistently throughout the manuscript? Please include details regarding the statistical analysis or data acquired from specific databases.

Figures and tables are appropriate, although table text is sometimes misaligned. Data has been correctly interpreted throughout the manuscript and discussion is thorough.

·        Are the conclusions consistent with the evidence and arguments presented?

Yes.

·        Please evaluate the ethics statements and data availability statements to ensure they are adequate.

Ethics statement are correct, but institutional review boards names and internal codes should be stated. There is not any data availability statement, and it should be included.

Specific comments

1.      Lines 36-37: Please state which variables were analyzed.

2.      Cited references should go before the punctuation marks. In the paper, references are situated after the punctuation marks (e.g. line 53). Please, put them before the marks.

3.      The first time that the term “ELSA-Brasil” appears in the text, a brief explanation of what ELSA means should be included to facilitate readers’ comprehension.

4.      Line 84: Missing full stop, there is a comma at the end of the sentence when it should be a full stop.

5.      Population and sample: Please include age median and standard deviation of the selected sample. A brief description of the context and/or background (cultural, socioeconomical…) of the participants should also be included. It helps to contextualize the data and the study.

6.      A study design subsection should be included within the Materials and Methods, in which an explanation of the research design is explained. Although you state that ELSA-Brasil methodology has been previously described, readers should be able to know what you have done and how you have designed the study without looking for a different paper.

7.      Have you carried out interviews in the study? If not, please change the word “interviewers” (line 97) for a different one, it could lead to a misunderstanding about the research. Interview is a qualitative research technique of data collection, and I cannot see anything qualitative in this study. You could use “researchers”, “evaluators” or any other synonym.

8.      What do you mean by “gauges”? (Line 97). It does not make sense. Maybe a translation mistake? I would go for the same word as in previous comment.

9.      Please, describe thoroughly the data collection procedure in subsection 2.2 (Data production). It is not clear how the questionnaires were applied, and what data has been analyzed.

10.   Is this a follow-up stage? Or a different stage after COVID-19 follow-up stage? It is difficult to understand in the paper. I believe it is an analysis of the differences of the follow-up versus the previous measures, but it is not clear.  

11.   In the subsection 2.4 (Assessment of self-reported SARV-CoV-2 infection), why have you used this criterion? Is there any reference that has used it previously?

12.   Line 142: Samples were also obtained. Samples of what? Blood?

13.   Line 154: Please, put it as a subsection (2.6.1).

14.   Specific practices to protect against Covid-19 (lines 155-161): Why have you used this criterion? Previous studies?

15.   Table 1: The text of the table in the LTPA data is not properly structured. You should adjust the lines of the table to line up male and female data.

16.   Table 2: Data is not lined up. Please correct it, same as Table 1.

17.   Lines 209-212: According to the data, it seems that the combination of physical activity + specific practices is the one that can reduce risk of infection. I think that stating “when compared to those who remained physically inactive” does not properly refer to the results and could lead to a misunderstanding.

18.   Lines 294-295: well defined cultural and socio-demographic differences: Which are those differences? See specific comment 5. Please correct differences, you have written diferences.

19.   I think you could include a paragraph in the discussion section talking about how the results may be generalized.

20.   Reference 21: there are cuts in the reference.

21.   Are there 2022 references that you could include to have more recent information?

22.   Please include data availability statement.

23.   Please include institutional review boards/ethics committees’ names and codes.

24.   Although you describe in the study the variables of obesity, hypertension and diabetes, I do not believe they add any relevant information or affect the results of the research. I think they could be excluded. If you believe they are important for the results, please include the reason in the manuscript. 

Round 2

Reviewer 2 Report

None

Author Response

Once again, thank you for your comments and suggestions.

Reviewer 4 Report

Dear authors,

Thank you for the corrections in your manuscript. I think there are still some issues to be changed in your manuscript in order to make it proper for publication:

1.      Line 70 and 72: You repeat “It is important to note…”. Please change it to a synonym.

2.      Population and sample: Description of context and/or background is still not provided (lines 90-91). You just included that there are cultural and sociodemographic differences. You should describe in 2-3 lines what are these differences (e.g. …well-defined cultural and sociodemographic differences, ranging from … to …).

3.      Please correct the English writing in lines 91-97. There are some grammar mistakes.

4.      Line 95: Volunteers or participants?

5.      Line 103: What is a longitudinal observational design? Please add a description of this research design.

6.      Line 113: Interviewers is still in the text. If you are not doing any interview, please remove this. It could lead to a confusion about the data collection.

7.      Lines 137-138: You did not answer my previous question about why you chose this criterion for reporting SARS-CoV2 infection. It is clear that you used the question “Did you have a positive test for COVID-19?”, but I want to know why you decided to use this question.

8.      Lines 169-177: Please refer to the publications that demonstrate that those practices can reduce COVID transmission. I cannot see any reference in the methods section, and it should be stated in this section.

9.      Lines 225-228 (specific comment 17 in previous review): I still don’t agree with this statement. Even if the group who remained inactive was the reference in your analysis, the results show that only a combination of physical activity and specific protection practices can reduce risk of infection. I believe that stating “those who remained physically active during the follow-up and carried out specific practices to protect against COVID-19 during the pandemic were less infected when compared to those who remained physically inactive” does not adjust to the results, since the people who remained active but did not carry out specific protection practices did not reduce risk of infection. Please, change this statement to something that adapts to the results.

10.   Please, check thoroughly the whole document for English mistakes, specially the newly added information.

Thank you for your time and effort to contribute to scientific literature. Best regards. 
